# The Influences of Soil and Meteorological Factors on the Growth and Fruit Quality of Chinese Jujube (*Ziziphus jujuba* Mill.)

**DOI:** 10.3390/plants12244107

**Published:** 2023-12-08

**Authors:** Yanjun Duan, Shuang Liu, Ying Zhu, Yongkang Wang, Fenfen Yan, Zhiguo Liu, Xiaoxin Shi, Ping Liu, Mengjun Liu

**Affiliations:** 1Research Center of Chinese Jujube, Hebei Agricultural University, Baoding 071001, China; duan13091259695@163.com (Y.D.); ls79820@163.com (S.L.); 17797964949@163.com (Y.Z.); jujubeliu@163.com (Z.L.); 2College of Horticulture, Hebei Agricultural University, Baoding 071001, China; 3Hebei Provincial Technology Research Institute of Jujube Industry, Baoding 071001, China; 4Pomology Institute, Shanxi Agricultural University, Jinzhong 030800, China; woyok63@126.com; 5College of Plant Science, Tarim University, Alar 843300, China; yanfening@163.com; 6Meteorological Bureau of Taigu District, Jinzhong 030800, China; sxtgqxj@126.com

**Keywords:** jujube, climatic adaptability, soil nutrients, fruit quality, redundancy analysis, principal component analysis

## Abstract

Chinese jujube (*Ziziphus jujuba* Mill.) is attracting more and more attention worldwide due to their tasty and nutritious fruit with extremely high contents of vitamin C (Vc) and soluble sugar. In order to find out the main factors that influence jujube growth and reproductive adaptability, the phenological periods, vegetative growths, fruiting abilities, and fruit qualities of eight newly released cultivars were compared and comprehensively analyzed in three representative ecological sites of the three main jujube-producing regions including Fuping (Hebei), Taigu (Shanxi), and Alar (Xinjiang) in China. Our results showed that the characteristics of jujube cultivars were significantly affected by soil and meteorological factors. The fruit number per bearing shoot was much more affected by temperature, light, and rainfall. The fruit number per bearing shoot, contents of soluble solids, and soluble sugar and Vc contents in fruits were influenced more by meteorological factors. The content of flavonoids was affected by both soil and meteorological factors. A principal component analysis (PCA) indicated that cultivars suitable for planting in Fuping (Hebei) were Yuhong and Lengbaiyu. Zaocuimi, Fucuimi, and Zaoqiuhong were suitable to be cultivated in Taigu (Shanxi), while Zaocuimi, Yuhong, Yulu, Luzao 2, and Yueguang behaved better in Alar (Xinjiang). This study provides insights of the environmental factors on jujube yield and quality and therefore provides references for highly efficient jujube cultivation.

## 1. Introduction

The growth and development of fruit trees are affected by genotype, cultivation management, and environmental conditions. There have been many reports on the influences of genotype and cultivation management. However, environmental conditions have a great impact on the growth of fruit trees, especially on their fruit quality. This impact is very complex and difficult to study, resulting in slow progress in related research, and many problems are still unclear.

Chinese jujube (*Ziziphus jujuba* Mill.), the most economically important member of the Rhamnaceae family, is one of the oldest cultivated fruit trees in the world [1]. It originates in the middle and lower reaches of the Yellow River, China, and has been introduced into at least 48 countries in all continents except Antarctica, becoming increasingly important, especially in arid and semi-arid regions [2], owing to its multiuse nutritious fruits [3,4,5]. Jujube fruits contain sugars, acids, triterpenoids, alkaloids, flavonoids, and cyclic adenosine monophosphate (cAMP), and they have higher antioxidant capacity [6,7,8]. With the continuous and rapid development of the jujube industry, table jujube becomes more and more popular owing to its better taste (crispy) and much higher vitamin C content than dehydrated jujube [9].

The ecological environment factors differ owing to the differences in the climates and soils of the cultivation regions. Cultivation practice showed that ecological differences had significant effects on the cultivar adaptability, growth and development, fruit yield, and quality. For example, temperature, humidity, and solar radiation can affect the morphological characteristics of fruits, such as germination, development, and ripening. In particular, the phenotypes of the fruits, such as the weight and color, and nutritional quality, such as soluble solids and organic acid contents, as well as pest and disease resistance, are significantly affected by the ecological environment [10,11,12]. So far, ecological effects on grape, wolfberry, and citrus varieties have been reported [13,14,15]. As for Chinese jujube, most of the previous cultivar evaluations are limited to the evaluation of different cultivars in a certain region or the investigation of the same jujube cultivar in different environments [16,17]. Previous studies showed that soil is the main factor affecting fruit tree cultivation. Soil nutrients directly affect the growth of trees, and appropriate soil nutrients can improve the nutritional quality of fruits. For example, organic matter, nitrogen, phosphorus, and potassium will affect the nutritional quality of jujube fruit, such as the flavonoid contents and soluble solid contents [18,19,20]. In addition, the characteristics of jujube fruit are significantly related to temperature [21]. High temperatures significantly affected the accumulation of sugar, the degradation of organic acid content, the accumulation of anthocyanins, and skin coloring in other fruit trees [22,23,24]. However, the specific effects of ecological conditions on the growth and development of different jujube cultivars remained unclear. Therefore, it is of practical significance for the high-quality development of the table jujube industry to select excellent table jujube cultivars that are suitable for cultivation in different regions, making reasonable use of unique environmental conditions, and providing region-dependent cultivars with distinctive advantages. 

In this study, meteorological factors and soil nutrients were collected and investigated in three major jujube-producing areas, including the Taihang Mountains (semi-arid mountainous region, Fuping, Hebei), Cold Loess Plateau (Taigu, Shanxi) and Gravel Gobi Desert (Alar, Xinjiang), and fruit samples were analyzed to investigate the correlation between different meteorological and soil factors and jujube fruit quality to select cultivars that are suitable for different regions and to provide theoretical and practical references for the ecological planning and cultivation of jujube. 

## 2. Results

### 2.1. Differences in Meteorological Factors and Soil Conditions among Test Sites in Different Ecological Regions

The variability of ecological factors such as geographic location, altitude, and latitude and longitude in the three test sites resulted in differences in temperature and humidity, sunshine hours and rainfall in each area. In general, June to September was the key period for fruit setting and development; the most significantly different meteorological factors in the three test sites are sunshine hours, rainfall amount, relative air humidity and the daily temperature range.

In 2021, the monthly average temperature of the three test sites was shown to be the highest in July, while the lowest average minimum temperature appeared in Taigu (Shanxi) (Figure 1). In 2022, Taigu (Shanxi) had the highest average monthly temperature in June, and Alar (Xinjiang) had the lowest average minimum temperature (Figure 2). In general, the average maximum temperature from June to September is higher in Alar (Xinjiang), while the average minimum temperature is higher in Fuping (Hebei). The rainfall decreased gradually from east to west (Fuping, Hebei–Taigu, Shanxi–Alar, Xinjiang), while the annual sunshine duration lengthened, and the daily temperature range increased. The relative humidity (RH) also showed a gradually decreasing trend from east to west. The relative humidity in Fuping (Hebei) was the highest, and the relative humidity in 2021 was higher than that in 2022. 

The indices of the soil nutrients differed with the test sites. The soil nutrient qualities in different cultivation areas were analyzed, and they are shown in Table 1. The soil pH values in Taigu (Shanxi) and Alar (Xinjiang) are 8.41 and 8.21, respectively, implying that they are alkaline soils, and the soil pH value in Fuping (Hebei) is 6.97, indicating that it is neutral soil. In Taigu (Shanxi), the contents of organic matter, total nitrogen, total potassium, alkali nitrogen, available potassium, exchangeable calcium, exchangeable magnesium, and available manganese in the soil were the highest. In Alar (Xinjiang), the total phosphorus content, electrical conductivity (EC) value, available iron, and available zinc content in the soil were highest. In general, the soil nutrients in Fuping (Hebei) were lower than those in the other two test sites. 

### 2.2. Influences of Environmental Conditions on Phenological Periods

The phenological period differed with regions and cultivars (Table 2). And these differences directly affect the marketing period in different regions. Generally speaking, jujube germinated and ripened earlier in Alar (Xinjiang), which suggests that it may be affected by temperature and light. 

In Fuping (Hebei), the bud sprouting time of the eight table jujube cultivars varied from 15 April to 28 April, of which Yueguang sprouted earliest, while Luzao 2 sprouted the latest. The crisp mature periods of the different cultivars started from 4 September (Yueguang) to 30 September (Lengbaiyu). The defoliation date was the earliest in Yueguang on 3 October and the latest in Lengbaiyu on 16 November. Consequently, the early ripening cultivar, Yueguang, has the shortest fruit development period (85 days) and annual growth period (171 days), while the late cultivar Lengbaiyu has the longest fruit growth period (111 days) and annual growth period (210 days). 

In Taigu (Shanxi), the start of bud sprouting of the eight tested jujube cultivars lasted only 4 days, from 24 April (Zaoqiuhong) to 28 April (Zaocuimi). The fruit crisp mature period of all cultivars lasted only 5 days, from 5 September to 10 September. The crisp mature period of Lengbaiyu started 20 days earlier than that in Fuping (Hebei) and 10 days earlier than that in Alar (Xinjiang). The defoliation period start date lasted for 18 days, from 20 October (Zaocuimi) to 17 November (Fucuimi). The fruit development period of the different cultivars varied from 89 to 94 days; the annual growing period of Zaocuimi is the shortest (175 days), while that of Fucuimi is the longest (196 days).

In Alar (Xinjiang), the bud sprouting start date of the eight table jujube cultivars lasted only 3 days, from 18 April (Yueguang) to 21 April (most of the other cultivars). The fruit crisp mature period of all cultivars lasted only 26 days, from 25 August to 20 September. Fucuimi had the shortest fruit development period (78 days), while Yuhong and Yulu had the longest fruit development periods (107 days). The defoliation period of all the tested cultivars is concentrated around 20 October. Most cultivars have an annual growth period of about 182 days.

### 2.3. Influences of Environmental Conditions on Branching Ability

Simplified cultivation is becoming more and more important as the labor cost is increasing. Low branching ability is a key of simplified cultivation. It can be seen that the branching ability varied greatly among cultivars in different test sites (Table 3). In total, the jujube cultivars showed the highest branching ability in Taigu (Shanxi), followed by Fuping (Hebei) and Alar (Xinjiang). This might result from the highest soil fertility in Taigu (Shanxi). The branching abilities of Yuhong, Zaoqiuhong, and Fucuimi were much higher in all three sites. These cultivars were easy to propagate and to establish canopies. The branching abilities of the Yueguang, Lengbaiyu, and Zaocuimi cultivars were much lower in all three sites, which means that they have great potential in simplified cultivation.

### 2.4. Influences of Environmental Conditions on Productivity

The fruit number per bearing shoot (FNPBS) reflects the productivity of the cultivars. The FNPBS varied significantly with regions. It can be found that the FNPBS of the same cultivar varies greatly among the test sites. According to the investigation in 2021, the averaged FNPBS of eight cultivars in Alar (Xinjiang) was higher than that of Fuping (Hebei), and the averaged FNPBS of the cultivars in Taigu (Shanxi) was the lowest, which may be related to the climatic conditions and tree nutrition statuses of the pilot flowering periods in different regions. In Alar (Xinjiang), the averaged FNPBS of the eight tested jujube cultivars was 2.1 times that in Fuping (Hebei) and 23.0 times that in Taigu (Shanxi) (Table 4). Zaocuimi showed higher FNPBS values in Fuping (Hebei) and Alar (Xinjiang) (6.1 and 13.9, respectively). 

The FNPBS varied significantly with the cultivars. The average FNPBS value of Zaocuimi is significantly higher than those of the other cultivars. Lengbaiyu has the lowest average FNPBS value.

The FNPBS also varied significantly with years. Compared with 2021, the FNPBS of the jujube cultivars in Fuping (Hebei) decreased in 2022, especially Luzao 2, which decreased by 80%. The FNPBS of the cultivars in Taigu (Shanxi) increased in 2022 compared to that in 2021, while in Alar (Xinjiang), the FNPBS difference between two years was small.

As for the fruit weight, it is mainly related to the cultivars themselves and the yield. The higher the FNPBS, the lower the fruit weight of a certain cultivar (Table 5). For example, the fruit weights of Zaoqiuhong in Fuping (Hebei) and Taigu (Shanxi) were larger than those in Alar (Xinjiang), the fruit weights of the other cultivars in Taigu (Shanxi) were larger, and the fruit weights in Alar (Xinjiang) were relatively small.

In addition, the fruit weights differed with regions. For example, the coefficient of variation of the fruit weight was large in different regions. The single fruit weights of Zaoqiuhong and Lengbaiyu in the three test sites were significantly different, which may be related to the yield, soil nutrients, and other climatic conditions.

### 2.5. Influences of Environmental Conditions on Fruit Quality

The soluble solid content, soluble sugar content, and titratable acid content of jujube fruit are important indexes to evaluate the taste, flavor, and nutritional quality of jujube fruit. The sugar/acid ratio (the ratio of soluble sugar to titratable acid) and the solid/acid ratio (the ratio of soluble solid to titratable acid) determines the taste of table jujube. The above indicators are not only affected by the characteristics of the cultivar itself, but are also easily affected by the ecological conditions.

The soluble solid content of jujube fruit differed with cultivars, regions, and years. In 2021, it can be seen that the soluble solid content of the jujube fruit in Alar (Xinjiang) was generally higher than that in the other two sites; it was 15.3% higher than that Fuping (Hebei) and 3.0% higher than that in Taigu (Shanxi). Except for Fucuimi and Lengbaiyu, the other six cultivars behaved differently in terms of the soluble solid content among the three test sites. The soluble solid content of Luzao 2 in Taigu (Shanxi) was 32.0%, and the soluble solid content of Zaocuimi in Alar (Xinjiang) was the highest. The soluble solid content of Yulu was higher in Fuping (Hebei) and Taigu (Shanxi) (24.9% and 25.2%, respectively) (Figure 3A). Compared with 2021, the soluble solid content of the jujube cultivars in 2022 had little difference between Fuping (Hebei) and Alar (Xinjiang), but in Taigu (Shanxi), the soluble solid contents of Luzao 2, Yueguang, and Zaoqiuhong showed a decreasing trend (Figure 4A).

The soluble sugar content of jujube fruits also behaved differently among cultivars, regions, and years. In 2021, the soluble sugar content in Alar (Xinjiang) was generally higher than that in Fuping (Hebei) and Taigu (Shanxi); it was 22.6% higher than that in Fuping (Hebei) and 4.7% higher than that in Taigu (Shanxi) (Figure 3B). Lengbaiyu and Yulu did not show significant differences in the soluble sugar contents among the three sites. The soluble sugar contents of Fucuimi, Luzao 2, Yuhong, and Yueguang were higher in Taigu (Shanxi) and Alar (Xinjiang), and the soluble sugar content of Zaocuimi was higher in Alar (Xinjiang). Compared with 2021, the soluble sugar content in Xinjiang had a great difference in two years and showed a decreasing trend. Particularly, the soluble sugar contents of Luzao 2, Yulu, Zaocuimi, and Zaoqiuhong in Alar (Xinjiang) decreased largely in 2022 (Figure 4B).

The titratable acid content in the jujube fruits was affected mainly by the cultivar, followed by the region and year. In 2021, there was little difference in the titratable acid content among the three sites, indicating that environmental changes had little effect on the titratable acid content. But there was a great difference among the cultivars in the same site (Figure 3C). The titratable acid content of Yulu was higher than that of the other cultivars and was 1.99% higher in Taigu (Shanxi). There was no significant difference in the titratable acid contents of Lengbaiyu, Luzao 2, and Yueguang among the three test sites. However, the titratable acid content of Yuhong was significantly different among the three sites, with the highest content in Alar (Xinjiang) (0.41%) and the lowest in Fuping (Hebei) (0.25%). Compared with 2021, the titratable acid content was significantly lower in Fuping (Hebei) in 2022. Among all of the tested cultivars, the titratable acid contents of Zaocuimi and Zaoqiuhong decreased greatly (Figure 4C).

The sugar/acid ratio and solid/acid ratio differed a lot with cultivar, region, and year. In 2021, the sugar/acid ratios of different jujube cultivars in Fuping (Hebei) varied from 12.1 to 79.7 with an average of 48.3, and the ratio of the total soluble solid to acid varied between 14.7 and 96.8 with an average of 57.2. The ratio of sugar to acid of the jujube cultivars in Taigu (Shanxi) was 11.8~85.9 with an average of 62.9, and the ratio of the total soluble solid to acid was 12.7~103.2 with an average of 71.7. In Alar (Xinjiang), the ratio of sugar to acid was 15.2~87.2 with an average of 57.6, and the ratio of total soluble solid to acid was 15.9~92.1 with an average of 64.4. The two ratios of Yulu were the lowest in the three test sites. There were no significant differences in the two ratios of Lengbaiyu, Yueguang, and Zaocuimi among the three test sites (Figure 3G,H). Compared with 2021, the sugar/acid ratio and solid/acid ratio of all jujube cultivars in Fuping (Hebei) increased in 2022, with no significant changes in Alar (Xinjiang) or Taigu (Shanxi) (Figure 4G,H).

The Vc contents in jujube fruits were affected by the cultivar, region, and year. In 2021, the average Vc contents of all tested cultivars in Alar (Xinjiang) were higher than those in Fuping (Hebei) and Taigu (Shanxi); they were 40.2% higher than Fuping (Hebei) and 0.9% higher than Taigu (Shanxi) (Figure 3D). There was no significant difference in the Vc contents of Luzao 2 and Zaoqiuhong in the three test sites. The Vc contents of Yulu, Lengbaiyu, Yuhong, and Fucuimi were significantly different among the three sites. Among them, the Vc contents of Yuhong and Yulu were the highest in Taigu (Shanxi) (451.5 mg/100 g and 401.3 mg/100 g, respectively), while the Vc contents of Lengbaiyu and Fucuimi were the highest in Alar (Xinjiang) (354.24 mg/100 g and 306.13 mg/100 g, respectively) (Figure 3D). Compared with 2021, the Vc content in the fruits was significantly lower in Fuping (Hebei) in 2022. For example, the Vc contents of Lengbaiyu and Yuhong decreased significantly in 2022 (Figure 4D). 

The total flavonoid content in the fruits differed largely among cultivars, regions, and years. In 2021, the total flavonoid content in Fuping (Hebei) was higher than that in Taigu (Shanxi) and Alar (Xinjiang); it was 85.3% higher than that in Taigu (Shanxi) and 25.2% higher than that in Alar (Xinjiang) (Figure 3E). The total flavone content of Yulu in Alar (Xinjiang) was higher (247.3 mg/100 g), but there was no significant difference between that in Fuping (Hebei) and Taigu (Shanxi). There were significant differences in the other jujube cultivars among the three sites, and they were all the highest in Fuping (Hebei) and the lowest in Taigu (Shanxi), indicating that different environments had great influences on the flavonoid contents (Figure 3E). In 2022, the flavonoid content was greatly increased in the three test sites (Figure 4E).

The total phenol contents of the fruits were also different among cultivars, regions, and years. In 2021, the total phenol content in Taigu (Shanxi) was higher than that in Fuping (Hebei) and Alar (Xinjiang); it was 12.4% higher than that in Fuping (Hebei) and 4.0% higher than that in Alar (Xinjiang) (Figure 3F). The total phenolic contents of Yuhong and Yulu varied significantly among the three test sites, with the highest content in Taigu (Shanxi). The total phenolic content of Lengbaiyu was the highest (577.04 mg/100 g) in Fuping (Hebei), and that of Zaocuimi was the highest (746.35 mg/100 g) in Alar (Xinjiang). The total phenolic contents of Fucuimi, Luzao 2, and Zaoqiuhong did not show significant differences in the three test sites. In 2022, totally, the total phenol content in the fruits showed a little increase (Figure 4F).

### 2.6. Relationship between Jujube Fruit Quality and Meteorological Factors

June to September is an important period for the flowering, fruiting, and development of jujube. The data of the monthly average temperature, monthly average maximum temperature, monthly average minimum temperature, monthly average rainfall, relative air humidity, and sunshine hours from June to September for 2021 and 2022 were selected for a redundancy analysis (RDA) (Figure 5).

The quadrant of the arrow indicates the positive and negative correlation between the meteorological factors and the sorting axes (X axes and Y axes). The longer the arrow line of the meteorological factor, the greater the correlation between the meteorological factor and fruit quality, and the greater the influence on the fruit quality. The angular cosine between the arrows indicates the magnitude of the two correlations.

In general, the FNPBS, soluble solid content, Vc content, flavonoid content, and fruit weight were closely related to meteorological factors, while the soluble sugar, titratable acid content, and total phenolic content were not closely related to meteorological factors (Figure 5).

The FNPBS was positively correlated with the average temperature, average maximum temperature, and sunshine duration, and it was negatively correlated with the relative humidity and rainfall. The contents of soluble solids, soluble sugars, and Vc were positively correlated with the daily temperature range and sunshine hours, and negatively correlated with the average minimum temperature, rainfall, and relative air humidity. The flavonoid content in the fruit was positively correlated with the average minimum temperature and average temperature, positively correlated with rainfall, and negatively correlated with sunshine hours. 

### 2.7. Relationship between Jujube Fruit Quality and Soil Conditions

By analyzing 14 soil nutrient factors and 10 indexes for fruit productivity and quality, the FNPBS, soluble solid content, Vc content, fruit weight, and flavonoid content were closely related to the soil nutrient factors, while the soluble sugar content, titratable acid content, and total phenol content were not closely related to the soil nutrient factors (Figure 6).

FNPBS was positively correlated with the available phosphorus, available iron, and EC in soil, and it was negatively correlated with the organic matter content, alkaline nitrogen content, and exchangeable calcium content. Among them, the FNPBS was significantly affected by the available iron and exchangeable calcium. The contents of soluble solids, soluble sugars, and Vc were positively correlated with the contents of total phosphorus, total potassium, available phosphorus, available potassium, pH, EC, exchangeable magnesium, and available manganese in the soil.

### 2.8. Selection of Cultivars Suitable for Different Regions

A principal component analysis (PCA) is one of the main methods used for the selection and breeding of agricultural and forestry cultivars. The 10 characters of the eight table jujube cultivars in three test sites were transformed into three principal components, and the principal components were extracted according to the eigenvalues and contribution rates of the principal components.

A total of three principal components were extracted, and the contribution rate of the first principal component was 27.50%. Among them, the most significant contributions were the ratio of sugar to acid, the ratio of the total soluble solid to acid, and the titratable acid content, which mainly reflected the taste expression and fruit size. The variance contribution rate of the second principal component was 25.90%, which integrated the information of soluble sugar, soluble solid, and Vc, mainly reflected the nutritional quality of the fruit. The contribution rate of the third principal component was 19.07%, which integrated the FNPBS and other information, including the flavone and total phenol contents, mainly reflecting the fruit yield and tree potential growth (Table 6).

A comprehensive evaluation score of the nutritional quality can be obtained using the principal component score function. The higher the evaluation score, the better the quality. A comprehensive evaluation showed that the overall performance of all of the cultivars in Alar (Xinjiang) was superior than that in Fuping (Hebei) and Taigu (Shanxi). In Fuping (Hebei), Yuhong and Lengbaiyu behaved better. In Taigu (Shanxi), Zaocuimi, Zaoqiuhong, and Fucuimi produced more high-quality fruits, while in Alar (Xinjiang), Zaocuimi, Yuhong, Yulu, Luzao 2, and Yueguang were the better choices (Table 7).

## 3. Discussion

Studies have shown that the improvement of yield and the quality of jujube fruit are related to cultivation and management techniques [25], climatic conditions [26], geographical environment conditions [27,28], and the soil nutrient content [29,30]. The blind introduction of new cultivars that are not suitable for planting in a region will not only cause a waste of resources, but also produce a low yield and poor-quality fruits, which will also have certain impacts on the production area and consumer market. According to the results of this study, the differences in the performances of the same jujube cultivars in the three test sites may be related to the soil nutrient content and climatic conditions.

This study found that the higher the content of organic matter, the more vigorous the growth of jujube trees, and the lower fertility of jujube trees, indicating that jujube trees have higher tolerances to poor soil. In addition, the fruit productivity was greatly affected by the available iron content in the soil. Iron is an essential trace mineral nutrient in plant growth, which affects the formation of the chloroplast structure in the plant leaves, and it is an important synthetic substance of many oxidoreductases. It plays an important role in respiration, photosynthesis, and nitrogen metabolism. The soluble solid content, soluble sugar content, and Vc content in fruits are positively and significantly correlated with sunshine hours; longer sunshine hours in a fruit’s rapid growing period are beneficial to the accumulation of sugar and organic matters. The water deficit enhanced the organic acid and soluble solid content in the fruits, but it had a slight adverse impact on the average fruit weight [31]. The sunshine hours from June to September in Fuping (Hebei) were relatively short, which may be one of the main reasons for the low soluble solid and soluble sugar contents of the jujube fruits. It was found that the contents of soluble solids, soluble sugars, and Vc were positively correlated with the contents of total potassium, available potassium, total phosphorus, and available phosphorus in soil. This is consistent with the findings in other citrus studies [32]. Potassium and phosphorus, as the essential elements, play important roles during plant growth and development [33]. Phosphorus is indispensable in plant photosynthesis and respiration, carbohydrate, lipid, and nitrogen metabolisms. Some results indicated that potassium and phosphorus fertilizers had important effects on the fruit weight [34,35]. Higher contents of phosphorus and potassium elements were beneficial to the increase in sugar and Vc contents in fruits. The contents of flavonoids in the fruit were negatively correlated with the organic matter content, total nitrogen content, total potassium content, alkali nitrogen, available potassium, pH, exchangeable magnesium, and available manganese in the soil. This indicates that the barren soil is conducive to the accumulation of secondary metabolite flavonoids. The branching ability varies greatly among cultivars in different test sites, which may be related to cultivar characteristics, tree vigor, climate conditions, etc.

The total flavonoid content in the fruits was positively correlated with the mean temperature degree, minimum temperature, and rainfall, and it was negatively correlated with sunshine hours, total nitrogen content, and fast-acting potassium content, probably due to a combination of environmental and meteorological factors that increased the activity of the related enzymes in the flavonoid synthesis pathway, thus promoting the synthesis and accumulation of flavonoids. It was hypothesized that the flavonoid content would increase significantly under adverse conditions [36]. Our result indicates that high humidity, low temperature, and low light conditions favor the accumulation of flavonoids. This is shown by the correlation of FNPBS with the mean temperature, mean maximum temperature, sunshine duration, and rainfall. Higher temperature and light are very favorable for fruit settings, and too much rainfall will cause the fall of flowers and fruits. Sufficient light and diurnal temperature differences favor the accumulation of sugar and Vc, while low temperatures and rainfall decrease the sugar and Vc accumulations. It is clear from the above that there are various factors affecting the growth and development of jujube trees and fruits. This is consistent with previous studies [37,38]. It is of great importance for the development of the jujube industry to comprehensively analyze the key meteorological factors and soil nutrients that affect the productivity and fruit quality, and screen out the improved cultivars that can adapt to specific environmental conditions.

## 4. Materials and Methods

### 4.1. Plant Materials

Three representative test sites, including Fuping in Hebei province (semi-arid Taihang Mountain), Taigu in Shanxi province (Loess Plateau), and Alar in Xinjiang province (Gravel Gobi desert area), were selected. Fuping, located in the semi-arid Taihang Mountain in Hebei province, belongs to a continental monsoon climate with an elevation of 331 m, an average annual temperature of 13.2 °C, an annual precipitation of 812.3 mm, and a frost-free period of 180 days. Taigu in Shanxi province has a warm temperate continental climate with an elevation of 850 m, an average annual temperature of 12.3 °C, a rainfall of 646.3 mm, and a frost-free period of 175 days. Alar in Xinjiang has a temperate continental climate with an elevation of 1001 m, an average annual temperature of 11.8 °C, an annual precipitation of 51.3 mm, and a frost-free period of 168~171 days.

The regional test cultivars include Fucuimi, Lengbaiyu, Luzao 2, Yuhong, Yulu, Yueguang, Zaocuimi, and Zaoqiuhong. The test was carried out in 2020 by top grafting each cultivar on 15~30 adult jujube trees with 5~6 main branches per tree. Jujube fruits were collected in 2021–2022 at the half-red maturity stage, and 30 representative fruits were collected from 5~6 trees of each cultivar and repeated three times.

### 4.2. Soil Nutrients

Soil samples were selected at 5 points in the southeast, northwest, and middle of each district pilot, and soil from the root distribution area (at the crown projection, at a depth of about 30 cm, avoiding the fertilization point) was fully mixed and divided into 3 parts. The soil samples were dried and smashed until a particle diameter of less than 2 mm was obtained. The nutrient test was conducted at Baoding Jiyi Test Service Corp.

The determination of soil pH refers to NY/T1121, 2-2006. The 10 g soil samples were extracted with 25 mL of pure water and analyzed using pH Meter (Shanghai Youke Instrument Co., Ltd., Shanghai, China).

The determination of soil organic matter refers to NY/T 1121.6-2006. The soil samples were oxidized by 0.4 mol/L of K_2_Cr_2_O_7_–H_2_SO_4_ solution under oil boiling at 180 °C for 5 min. The remaining K_2_Cr_2_O_7_ was titrated with 0.2 mol/L FeSO_4_, and the organic matter content was calculated from the amount of K_2_Cr_2_O_7_ consumed.

The determination of soil nitrogen and alkaline nitrogen refers to LY/T 1228-2015. To determine soil nitrogen contents, 1 g soil samples was mixed with 2 g accelerator (K_2_SO_4_:CuSO_4_·5H_2_O = 10:1) with the addition of 5 mL concentrated H_2_SO_4_ and boiled for 1 h after the mixture was grey. Then, the mixture was cooled and analyzed using the Flow Injection Analyzer (FIA-6000+, Yitian Instrument, Beijing, China). To determine the soil alkaline nitrogen, 2 g soil samples was laid in the outer layer of the diffusion dish, and 3 mL of 20 g/L H_3_BO_3_ was poured in the inner layer. An amount of 10 mL of 1.8 mol/L NaOH was added in the outer layer and sealed, and the dish was laid in the incubator for 24 h under 40 °C; the dish was slightly shaken once every 8 h. The inner solution was titrated with 0.01 mol/L HCl.

The determination of the soil’s total phosphorus refers to NY/T 88-1988. An amount of 0.25 g soil samples was added in nickel crucible and moistened with several drops of anhydrous ethanol. And 2 g NaOH was laid on the samples. The temperature was improved from 400 °C to 720 °C and maintained for 15 min. An amount of 10 mL of distilled water was added at 80 °C and transferred to a volumetric flask with 10 mL of 3 mol/L H_2_SO_4_ solution and fixed the volume to 100 mL with distilled water. An amount of 2 mL of the test solution was sucked into a 50 mL volumetric flask, 2~3 drops of 2,4-dinitrophenol were added, and it was adjusted to a slightly yellow color with 10% Na_2_CO_3_ solution; 5 mL of molybdenum antimony antichromogenic agent was added and shaken well and the volume was fixed to 50 mL. About 30 min later, samples were analyzed using a UV spectrophotometer (Jinan Precision Electronic Technology Co., Ltd., Jinan, China).

The determination of the soil’s total potassium refers to NY/T 87-1988. An amount of 0.1 g soil samples was laid in teflon crucible and mixed with 3 mL HNO_3_ and 0.5 mL HClO₄. The mixture was heated into paste and mixed with 5 mL HF and 0.5 mL HClO₄ until the white smoke disappeared. The mixture was cooled, and 10 mL 3 mol/L HCl was added. Then, 2 mL of 2% H_3_BO_3_ was added and fixed to 100 mL with deionized water, and the mixture was analyzed using the Flame Photometer (6400A Shanghai Xinyi Precision Instrument Co., Ltd., Shanghai, China).

The determination of soil effective phosphorus refers to NY/T 1121.7-2014. An amount of 2.5 g soil samples was extracted with 50 mL of 0.5 mol/L NaHCO₃ and shaken for 30 min under 25 °C. The mixture was filtered, and 10 mL solution was transferred into a volumetric flask with the addition of 5 mL molybdenum antimony antichromogenic agent, and the volume was fixed to 50 mL with distilled water. About 30 min later, solution was analyzed with a UV spectrophotometer (Jinan Precision Electronic Technology Co., Ltd., Jinan, China).

The determination of the soil’s available potassium refers to NY/T 889-2004. An amount of 2.5 g soil samples was extracted in 1 mol/L of ammonium acetate, shaken for 30 min, and analyzed using the Flame Photometer (6400A Shanghai Xinyi Precision Instrument Co., Ltd., Shanghai, China).

The determination of soil’s exchangeable calcium and magnesium refers to NY/T1121.12-2006. An amount of 2 g soil samples was extracted in 60 mL ammonium acetate, and the mixture was centrifuged for 3~5 min at 3000~4000 r/min. The supernatant was collected in 250 mL volumetric flask 3 times. An amount of 20 mL supernatant was mixed with 5 mL SrCl_2_, and the volume was fixed to 50 mL and analyzed via Inductively Coupled Plasma Optical Emission Spectrometry (NCS Testing Technology Co., Ltd., Beijing, China).

The analyses of available iron, available manganese, and available zinc refer to NY/T 890-2004. An amount of 10 g soil samples was extracted in diethylenetriaminepentaacetic acid (DTPA) for 2 h under 25 °C and analyzed using Inductively Coupled Plasma Optical Emission Spectrometry (NCS Testing Technology Co., Ltd., Beijing, China).

The analysis of electrical conductivity refers to HJ802-2016. An amount of 20 g soil sample was mixed with 100 mL deionized water and shaken for 30 min under 20 ± 1 °C. About 30 min later, the solution was detected with a Conductivity Meter produced by Ohus Instruments (Changzhou) Co., Ltd., Changzhou, China.

### 4.3. Meteorological Data Acquisition

The meteorological data are taken from meteorological observation stations in each county and district, mainly including average temperature, minimum temperature, maximum temperature, temperature difference, sunshine hours, precipitation, and average relative humidity.

### 4.4. Determination of Growth and Fruiting Traits of Different Cultivars

A phenological investigation (bud sprouting, early blooming, full blooming, fruit crisp mature, fruit full mature, and defoliation stage), and growth index determination (branching ability) were carried out in 2021. Four to five trees of each cultivar were investigated, and 10 representative biennial branches were selected (with relatively consistent growth, with each branch being about 80–100 cm long, 6–10 nodes) for the investigation. The number of new shoots sprouting on the primary and secondary branches and the number of nodes on the primary and secondary branches were investigated in 18–19 May 2021, respectively. The branching ability was calculated (branching ability = the number of new shoots on all primary and secondary branches/the number of primary and secondary branches × 100%).

Economic traits including the fruit number per bearing shoot and the average fruit weight were investigated. All of the above-mentioned investigations referred to the Descriptors and Data Standard for Jujube (*Ziziphus jujuba* Mill.) [39].

### 4.5. Determination of Nutritional Quality of Jujube Fruits of Different Cultivars

Thirty representative fruits were collected from different directions of crowns of 5~6 trees, and the average weight of the fruit was measured and calculated. The selected fruits were pitted, mixed, and powdered in liquid nitrogen. The determination of the soluble solid content, soluble sugar content, titratable acid content, and vitamin C content referred to the refractometer method [40], the colorimetric method with 3,5-dinitrosalicylic acid [41], acid–base titration [42], and 2,6-dichloroindophenol titration [43], respectively. All of the tests were repeated 3 times.

### 4.6. Statistical Evaluation

Excel 2016 was used to process the data. A principal component analysis and a one-way analysis of variance (ANOVA) test for the data were conducted, and the significance differences among them were compared with a Duncan’s test (*p* < 0.05) using SPSS 26.0 (SPSS, Inc., Chicago, IL, USA). Canoco 5.0 was used for a redundancy analysis (RDA) to determine the relationship between the fruit quality and meteorological factors and between the fruit quality and soil conditions, respectively, and GraphPad Prism 8.0 software was used to map and analyze the data. Principal component analysis (PCA) was employed to select cultivars suitable for different regions.

## 5. Conclusions

In this study, we investigated the differences in the tree vigor and fruit quality among different regions and cultivars. A higher average temperature, sufficient light, and less rainfall are conducive to fruit yield. A longer sunshine duration, higher temperature, less rainfall, and temperature differences between the day and night are conducive to the accumulation of the soluble solid content, soluble sugar content, and Vc content in fruit. High rainfall, short sunshine hours, and poor soil conditions are more conducive to the accumulation of secondary metabolite flavonoids.

In view of the productivity and fruit quality, Yuhong behaved better in Fuping (Hebei), and Zaocuimi, Zaoqiuhong, and Fucuimi were suitable to be cultivated in Taigu (Shanxi). In Alar (Xinjiang), more cultivars such as Zaocuimi, Yuhong, Yulu, Luzao 2, and Yueguang showed promising futures.

## Figures and Tables

**Figure 1 plants-12-04107-f001:**
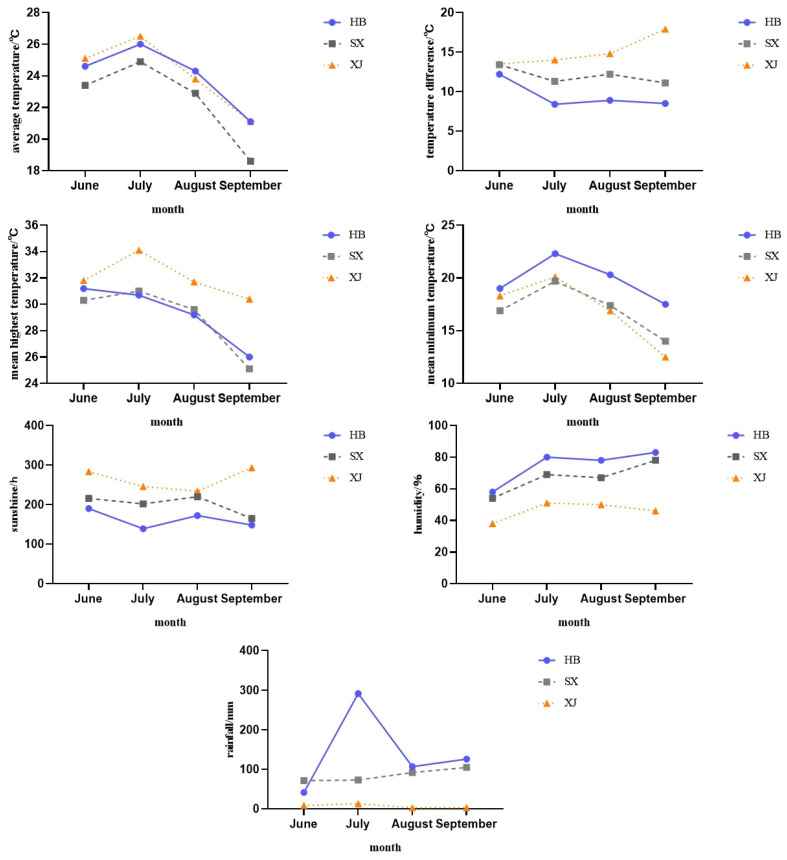
Meteorological factors in different test sites in 2021. Note: HB—Fuping (Hebei), SX—Taigu (Shanxi), XJ—Alar (Xinjiang). Same below.

**Figure 2 plants-12-04107-f002:**
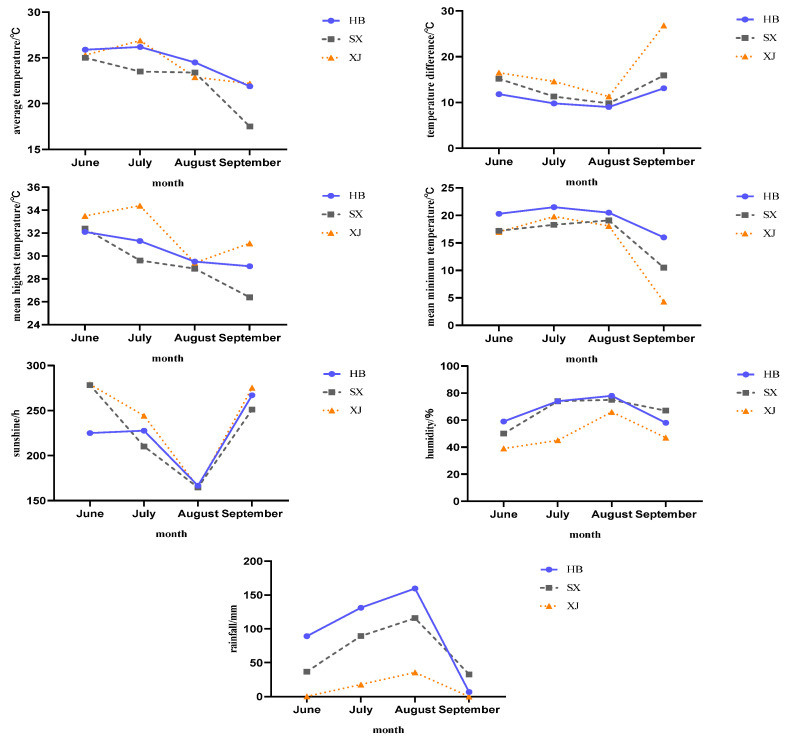
Meteorological factors in different test sites in 2022.

**Figure 3 plants-12-04107-f003:**
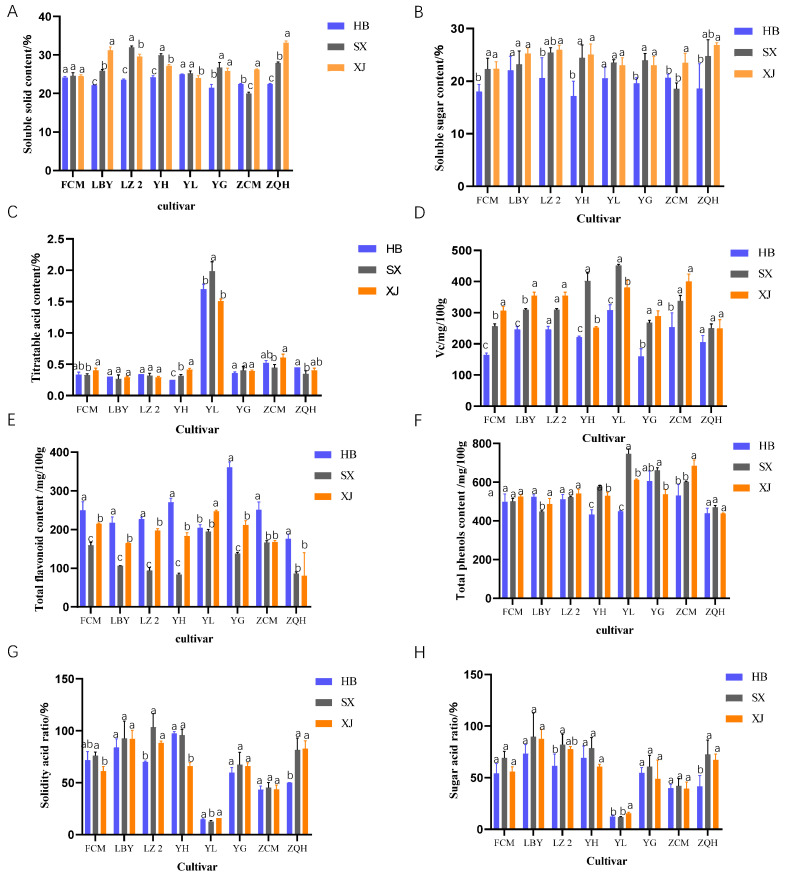
Nutritional quality of jujube cultivars in different test sites in 2021. ((**A**): soluble solid content; (**B**): soluble sugar content; (**C**): titratable acid content; (**D**): Vc content; (**E**): total flavonoid content; (**F**): total phenols content; (**G**): solidity acid ratio; (**H**): sugar acid ratio). Note: HB: Fuping (Hebei), SX: Taigu (Shanxi), XJ: Alar (Xinjiang). FCM (Fucuimi), LBY (Lengbaiyu), LZ 2 (Luzao 2), YH (Yuhong), YL (Yulu), YG (Yueguang), ZCM (Zaocuimi), and ZQH (Zaoqiuhong). Bar charts of different colors represent different test sites. Different letters represent significant differences among regions in the same cultivar (*p* ≤ 0.05).

**Figure 4 plants-12-04107-f004:**
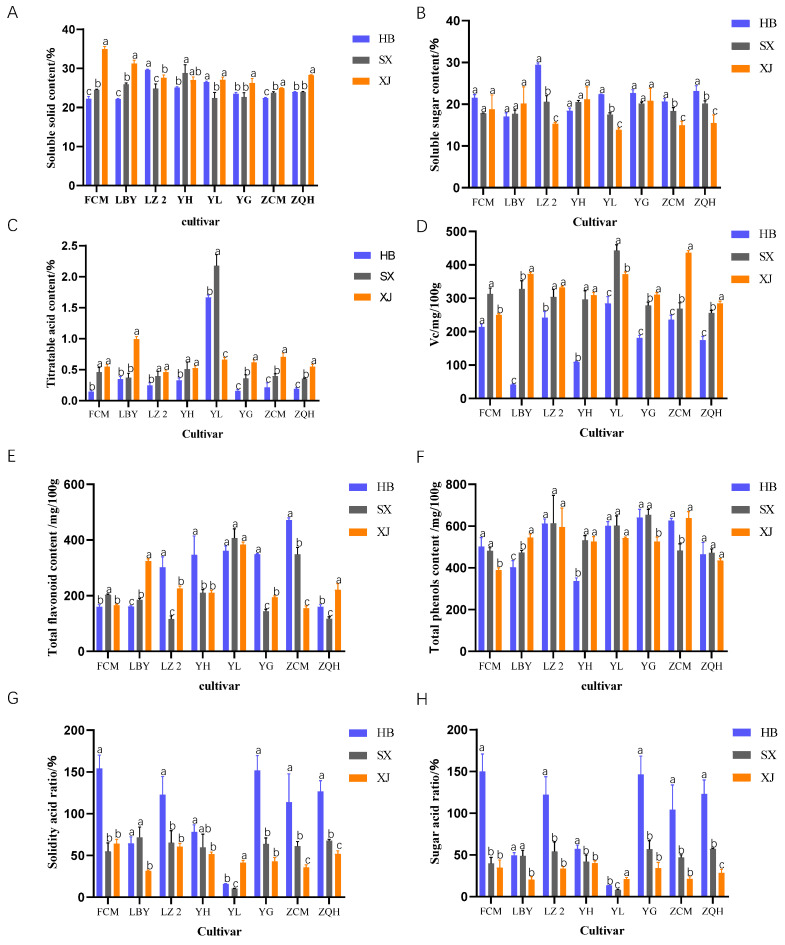
Nutritional quality of jujube cultivars in different test sites in 2022. ((**A**): soluble solid content; (**B**): soluble sugar content; (**C**): titratable acid content; (**D**): Vc content; (**E**): total flavonoid content; (**F**): total phenols content; (**G**): solidity acid ratio; (**H**): sugar acid ratio). Different letters represent significant differences among regions in the same cultivar (*p* ≤ 0.05).

**Figure 5 plants-12-04107-f005:**
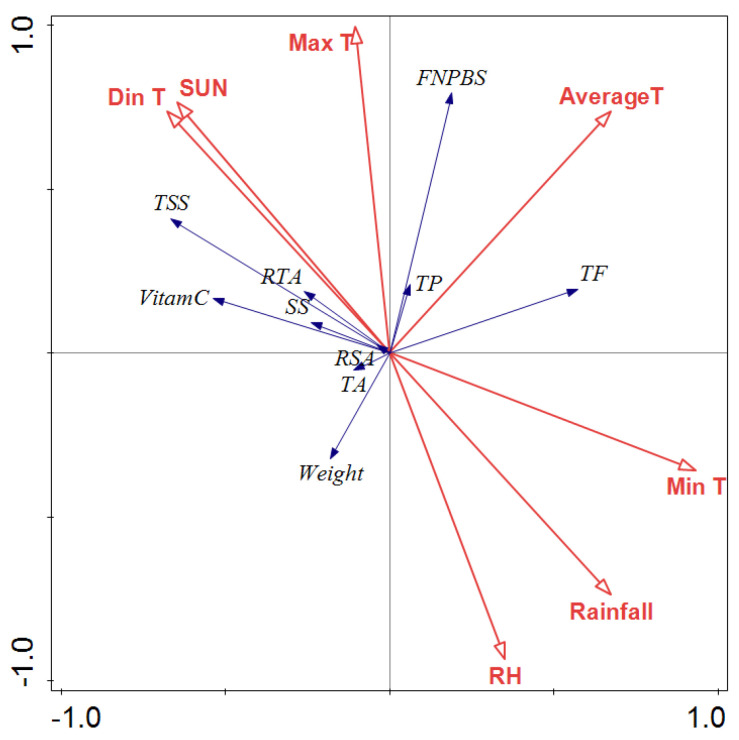
Relationship between jujube fruit characteristics and climate variables analyzed using CANOCO (5.12) software RDA. Note: AverageT—average temperature; Max T—mean maximum air temperature; Min T—mean minimum air temperature; Din T—temperature difference; RH—relative humidity; SUN—sunshine duration; TSS—total soluble solid; TA—titratable acid; SS—soluble sugar; RSA—ratio of total soluble solid to acid; RTA—ratio of sugar to acid; FNPBS—fruit number per bearing shoot; TP—total phenols; TF—total flavonoids; Weight—fruit weight.

**Figure 6 plants-12-04107-f006:**
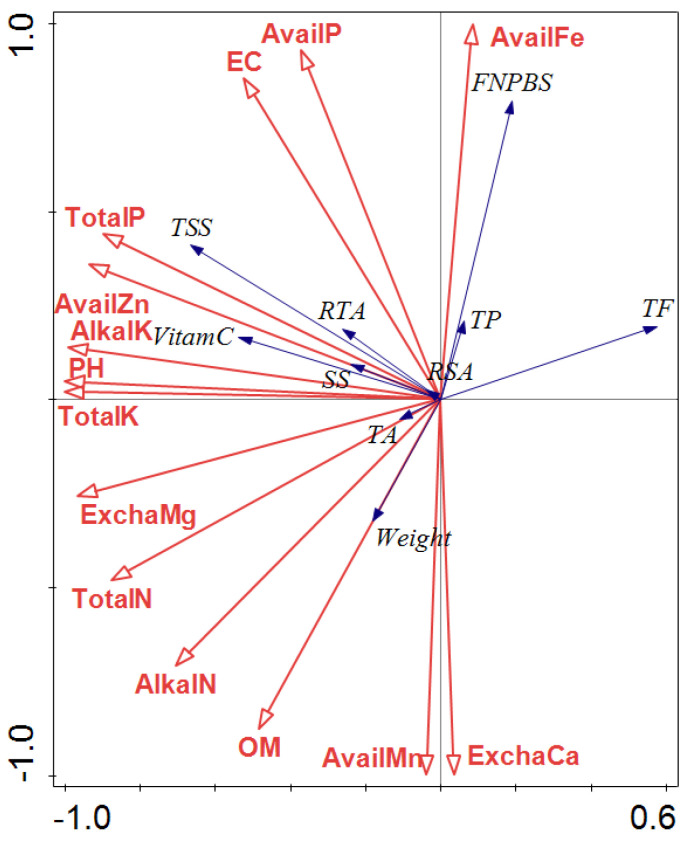
Relationship between jujube fruit traits and soil environmental variables analyzed using CANOCO (5.12) software RDA. Note: OM—organic matter; TotalN—total nitrogen; TotalP—total phosphorus; TotalK—total potassium; AlkalN—alkaline nitrogen; AvailP—available phosphorus; AlkalK—available potassium; EC—electrical conductivity; ExchaCa—exchangeable calcium; ExchaMg—exchangeable magnesium; AvailFe—available iron; AvailMn—available manganese; AvailZn—available zinc; TSS—total soluble solid; TA—titratable acid; VitamC—vitamin C; SS—soluble sugar; RSA—ratio of total soluble solid to acid; RTA—ratio of sugar to acid; FNPBS—fruit number per bearing shoot; TP—total phenols; TF—total flavonoids; Weight—fruit weight.

**Table 1 plants-12-04107-t001:** Comparison of soil composition in different test sites.

Items	Fuping (Hebei)	Taigu (Shanxi)	Alar (Xinjiang)
Organic matter (g·kg^−1^)	4.22 ± 0.18 b	10.03 ± 0.13 a	2.72 ± 0.17 c
Total nitrogen (g·kg^−1^)	0.27 ± 0.01 c	0.52 ± 0.01 a	0.36 ± 0.02 b
Total phosphorus (g·kg^−1^)	0.37 ± 0.02 c	0.60 ± 0.01 b	0.69 ± 0.02 a
Total potassium (g·kg^−1^)	8.55 ± 0.18 c	20.29 ± 0.05 a	18.33 ± 0.07 b
Alkali nitrogen (mg·kg^−1^)	20.83 ± 0.22 c	28.80 ± 0.17 a	21.60 ± 0.23 b
Available phosphorus (mg·kg^−1^)	2.02 ± 0.20 c	3.63 ± 0.20 b	45.65 ± 0.21 a
Available potassium (mg·kg^−1^)	27.52 ± 0.17 c	133.89 ± 0.34 a	129.43 ± 0.51 b
pH	6.97 ± 0.27 c	8.41 ± 0.31 a	8.21 ± 0.35 b
EC (mScm^−1^)	0.04 ± 0.01 b	0.15 ± 0.02 b	0.55 ± 0.21 a
Exchangeable calcium (mg·kg^−1^)	1273.50 ± 17.99 b	1670.30 ± 19.03 a	402.70 ± 17.24 c
Exchangeable magnesium (mg·kg^−1^)	119.50 ± 18.59 c	227.00 ± 20.28 a	181.40 ± 18.31 b
Available iron (mg·kg^−1^)	14.65 ± 0.43 b	9.46 ± 0.51 c	21.78 ± 0.43 a
Available manganese (mg·kg^−1^)	5.21 ± 0.20 b	6.45 ± 0.36 a	3.18 ± 0.30 c
Available zinc (mg·kg^−1^)	0.33 ± 0.08 c	0.65 ± 0.20 b	0.73 ± 0.07 a

Note: Different letters in the table indicate significant differences among cultivars in the same region at 0.05 level.

**Table 2 plants-12-04107-t002:** The phenological period of 8 table cultivars in different test sites.

Region	Cultivars	Bud Sprouting (Month/Date)	Early Flowering (Month/Date)	Full Flowering (Month/Date)	Fruit Crisp Mature (Month/Date)	Fruit Full Mature (Month/Date)	Defoliation (Month/Date)	Fruit Development Duration (Days)	Annual Growth Duration (Days)
Fuping (Hebei)	Fucuimi	4/25	6/5	6/11	9/13	9/25	10/21	94	179
	Lengbaiyu	4/20	6/7	6/11	9/30	10/14	11/6	111	210
	Luzao 2	4/28	6/5	6/11	9/13	9/25	10/29	94	181
	Yueguang	4/15	6/7	6/11	9/4	9/17	10/3	85	171
	Yuhong	4/22	6/5	6/11	9/13	10/19	10/29	94	190
	Yulu	4/22	6/2	6/6	9/13	9/22	10/29	99	190
	Zaocuimi	4/25	6/2	6/6	9/11	9/20	10/21	98	179
	Zaoqiuhong	4/27	6/7	6/15	9/13	9/20	10/26	90	182
Taigu (Shanxi)	Fucuimi	4/25	6/4	6/9	9/10	9/18	11/7	93	196
	Lengbaiyu	4/25	6/4	6/12	9/10	9/28	10/30	90	188
	Luzao 2	4/25	6/4	6/8	9/10	9/20	10/25	94	183
	Yueguang	4/25	6/4	6/8	9/5	9/14	10/22	89	180
	Yuhong	4/26	6/2	6/8	9/5	9/20	10/28	89	185
	Yulu	4/25	5/31	6/6	9/6	9/16	11/1	92	190
	Zaocuimi	4/28	6/7	6/25	9/8	9/16	10/20	89	175
	Zaoqiuhong	4/24	6/3	6/8	9/8	9/15	10/26	92	185
Alar (Xinjiang)	Fucuimi	4/21	5/15	6/15	9/1	9/15	10/20	78	182
	Lengbaiyu	4/21	5/18	5/26	9/20	9/25	10/20	86	182
	Luzao 2	4/21	5/17	5/25	9/15	9/20	10/20	87	182
	Yueguang	4/18	5/15	6/12	9/5	9/15	10/20	85	185
	Yuhong	4/21	5/19	5/25	9/10	9/20	10/20	107	182
	Yulu	4/21	5/20	5/25	9/10	9/20	10/20	107	182
	Zaocuimi	4/20	5/15	6/10	8/31	9/12	10/20	83	183
	Zaoqiuhong	4/21	5/15	6/15	8/25	9/18	10/20	92	182

**Table 3 plants-12-04107-t003:** Branching abilities of different cultivars in different test sites (unit, %).

Cultivars	Fuping (Hebei)	Taigu (Shanxi)	Alar (Xinjiang)	Average ^2^	Standard Deviation ^2^	Coefficient of Variation ^2^ (%)
Yulu	6.2 ± 0.33 a	16.9 ± 0.25 d	2.1 ± 0.24 d	8.40	6.24	74.28
Fucuimi	5.4 ± 0.29 ab	21.2 ± 0.51 c	2.8 ± 0.25 cd	9.80	8.13	82.97
Yuhong	5.4 ± 0.49 ab	36.6 ± 0.45 a	3.0 ± 0.41 bc	15.00	15.3	102.03
Luzao 2	5.0 ± 0.29 b	11.7 ± 0.45 e	4.0 ± 0.29 a	6.90	3.42	49.54
Zaoqiuhong	3.5 ± 0.45 c	24.9 ± 0.46 b	3.8 ± 0.47 ab	10.73	10.02	93.34
Zaocuimi	2.4 ± 0.45 d	3.3 ± 0.45 g	0.7 ± 0.09 e	2.13	1.08	50.53
Lengbaiyu	1.0 ± 0.29 e	3.0 ± 0.37 g	3.7 ± 0.54 ab	2.57	1.14	44.57
Yueguang	0.2 ± 0.04 f	5.3 ± 0.45 f	0 e	1.83	2.45	133.78
Average ^1^	3.6	15.4	2.5	/	/	/
Standard deviation ^1^	2.1	11.1	1.4	/	/	/
Coefficient of variation ^1^ (%)	57.5	72.5	55.2	/	/	/

Note: In the statistical parameters, superscript ^1^ represents an analysis among different cultivars in the same region, while superscript ^2^ represents an analysis among different regions in the same cultivar. Different letters in the table indicate significant differences among cultivars in the same region at 0.05 level.

**Table 4 plants-12-04107-t004:** Fruit number per bearing shoot of jujube cultivars in different test sites in 2021 and 2022 (unit: number).

Year	Cultivars	Fuping (Hebei)	Taigu (Shanxi)	Alar (Xinjiang)	Average ^2^	Standard Deviation ^2^	Coefficient of Variation ^2^ (%)
2021	Fucuimi	1.6 ± 0.37 cd	0.2 ± 0.08 bc	3.9 ± 0.29 bc	1.90	1.53	80.28
	Lengbaiyu	1.1 ± 0.22 d	0.1 ± 0.02 cd	2.0 ± 0.29 e	1.07	0.78	72.75
	Luzao 2	2.7 ± 0.16 b	0.1 ± 0.01 cd	2.9 ± 0.37 cd	1.90	1.28	67.13
	Yuhong	2.2 ± 0.22 bc	0.1 ± 0.02 cd	3.3 ± 0.29 cd	1.87	1.33	71.12
	Yulu	0.6 ± 0.22 de	0.7 ± 0.14 a	4.3 ± 0.54 b	1.87	1.72	92.20
	Yueguang	1.8 ± 0.37 c	0.03 ± 0.01 d	3.8 ± 0.29 bc	1.88	1.54	82.06
	Zaocuimi	6.1 ± 0.37 a	0.3 ± 0.09 b	13.9 ± 0.37 a	6.77	5.57	82.35
	Zaoqiuhong	1.8 ± 0.41 c	0.1 ± 0.05 cd	2.5 ± 0.29 de	1.47	1.01	68.71
	Average ^1^	2.2	0.2	4.6	/	/	/
	Standard deviation ^1^	1.6	0.2	3.6	/	/	/
	Coefficient of variation ^1^ (%)	70.5	99.5	78.6	/	/	/
2022	Fucuimi	2.1 ± 0.22 a	1.0 ± 0.37 b	4.2 ± 0.45 b	2.43	1.33	54.55
	Lengbaiyu	1.4 ± 0.29 bc	0.3 ± 0.11 c	0.9 ± 0.29 d	0.87	0.45	51.89
	Luzao 2	1.5 ± 0.29 bc	0.4 ± 0.09 c	2.5 ± 0.16 c	1.47	0.86	58.48
	Yuhong	2.1 ± 0.29 a	1.0 ± 0.51 b	2.6 ± 0.37 c	1.90	0.67	35.18
	Yulu	1.2 ± 0.37 bc	0.3 ± 0.07 c	3.2 ± 0.22 c	1.57	1.21	77.36
	Yueguang	1.7 ± 0.36 ab	0 c	4.7 ± 0.37 b	2.13	1.94	91.08
	Zaocuimi	1.2 ± 0.22 bc	1.6 ± 0.22 a	13.7 ± 0.29 a	5.50	5.80	105.46
	Zaoqiuhong	1.1 ± 0.09 c	1.3 ± 0.09 ab	2.6 ± 0.08 c	1.67	0.66	39.90
	Average ^1^	1.5	0.7	4.3	/	/	/
	Standard deviation ^1^	0.4	0.5	3.7	/	/	/
	Coefficient of variation ^1^ (%)	24.1	71.7	86.4	/	/	/

Note: In the statistical parameters, superscript ^1^ represents an analysis among different cultivars in the same region, while superscript ^2^ represents an analysis among different regions in the same cultivar. Different letters in the table indicate significant differences among cultivars in the same region at 0.05 level.

**Table 5 plants-12-04107-t005:** Fruit weights of jujube cultivars in different test sites in 2021 (unit: g).

Cultivars	Fuping (Hebei)	Taigu (Shanxi)	Alar (Xinjiang)	Average ^2^	Standard Deviation ^2^	Coefficient of Variation ^2^ (%)
Fucuimi	12.9 ± 0.29 c	13.2 ± 0.62 d	9.4 ± 0.45 d	11.83	1.72	14.58
Lengbaiyu	16.7 ± 0.37 b	22.3 ± 0.37 b	15.6 ± 0.29 a	18.20	2.93	16.12
Luzao 2	16.9 ± 0.29 b	18.6 ± 0.29 c	13.4 ± 0.45 b	16.30	2.16	13.28
Yuhong	12.1 ± 0.45 c	12.3 ± 0.29 de	11.9 ± 0.37 c	12.10	0.16	1.35
Yulu	7.2 ± 0.37 e	9.4 ± 0.45 f	5.9 ± 0.37 f	7.50	1.44	19.26
Yueguang	10.0 ± 0.29 d	11.9 ± 0.45 e	7.6 ± 0.37 e	9.83	1.76	17.89
Zaocuimi	9.6 ± 0.37 d	6.7 ± 0.29 g	7.9 ± 0.29 e	8.07	1.19	14.75
Zaoqiuhong	22.5 ± 0.45 a	23.5 ± 0.51 a	15.4 ± 0.28 a	20.47	3.61	17.62
Average ^1^	12.8	14.7	10.9	/	/	/
Standard deviation ^1^	5.5	5.7	3.5	/	/	/
Coefficient of variation ^1^ (%)	42.7	38.6	31.9	/	/	/

Note: In the statistical parameters, superscript ^1^ represents an analysis among different cultivars in the same region, while superscript ^2^ represents an analysis among different regions in the same cultivar. Different letters in the table indicate significant differences among cultivars in the same region at 0.05 level.

**Table 6 plants-12-04107-t006:** Principal component analysis of jujube quality.

Items	Eigenvector		
	F1	F2	F3
Soluble sugar content	0.04	0.90	0.05
Vc content	0.00	0.76	0.21
Titratable acid content	−0.92	0.24	0.03
Soluble solid content	0.03	0.77	−0.20
The ratio of total soluble solid	0.93	0.20	−0.06
The ratio of sugar to acid	0.94	0.18	−0.19
Fruit weight	0.01	−0.34	−0.72
FNPBS	0.10	0.21	0.68
Total flavonoid content	0.38	−0.57	0.61
Total phenol content	−0.03	−0.08	0.65
Eigenvalues	2.75	2.59	1.91
Variance account/%	27.50	25.90	19.07
Total account/%	27.50	53.39	72.47
Weight coefficient	0.38	0.36	0.26

**Table 7 plants-12-04107-t007:** Comprehensive evaluation of jujube quality in different regions.

Region	Cultivars	Comprehensive Evaluation				
		F1	F2	F3	F	Comprehensive Ranking
Fuping (Hebei)	Fucuimi	2.25	−3.44	0.00	−0.38	17
	Lengbaiyu	1.23	−0.90	−0.99	−0.12	14
	Luzao 2	−0.64	−3.10	−0.55	−1.56	24
	Yuhong	0.08	1.04	1.44	0.78	6
	Yulu	−0.21	−0.49	0.48	−0.13	15
	Yueguang	−0.91	−2.40	1.65	−1.20	23
	Zaocuimi	−1.17	−1.54	0.77	−0.79	18
	Zaoqiuhong	−0.88	−1.38	0.10	−0.80	19
Taigu (Shanxi)	Fucuimi	1.63	−0.18	−0.57	0.41	9
	Lengbaiyu	−0.63	−1.56	−1.44	−1.17	22
	Luzao 2	−1.21	0.44	−4.06	−0.30	16
	Yuhong	−0.50	0.71	−0.43	0.06	13
	Yulu	−3.68	1.33	−0.89	−1.15	21
	Yueguang	−2.73	0.42	−0.34	−0.97	19
	Zaocuimi	3.64	0.84	−0.04	1.67	2
	Zaoqiuhong	1.93	0.67	−1.33	0.62	7
Alar (Xinjiang)	Fucuimi	−0.25	−0.83	2.36	0.23	12
	Lengbaiyu	−0.28	0.36	0.92	0.26	11
	Luzao 2	0.22	2.00	−0.01	0.80	5
	Yuhong	0.30	0.69	1.96	0.88	3
	Yulu	−1.31	2.47	1.66	0.82	4
	Yueguang	−0.52	1.62	0.45	0.50	8
	Zaocuimi	2.73	2.08	0.26	1.85	1
	Zaoqiuhong	0.89	1.16	−1.40	0.39	10

## Data Availability

Data are contained within the article.

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
