# Peer review of "The Influences of Soil and Meteorological Factors on the Growth and Fruit Quality of Chinese Jujube (Ziziphus jujuba Mill.)"

_plants, 2023, doi:10.3390/plants12244107_

Round 1
Reviewer 1 Report
Comments and Suggestions for Authors
1) The text should be editorially improved. Latin names should be written in italics, e.g. line 15 and other places in the text. The space between letters and special characters is often not preserved, e.g. lines 16, 24, 80, 204 and other places in the text.
2) The scientific notation of the units is as follows: g kg-1 not g/kg.
3) There is no fruit weight in 2022, why?
4) There are no full statistics in Table 4. There is a coefficient of variation, but how to compare varieties?
5) I suggest unifying the terms for the location, currently the names are double, and additionally they are used interchangeably, for example Fuping once and then Hebei (line 141-159). This will make it easier for the reader to perceive the data.
6) The description of Fig. 3 should be complete and clearer. Mark clearly what is a location and what is a variety. It is necessary to explain the letters on the chart and the error bars. Figures can be presented in color. There is also no explanation of how to compare the results. Do we assess differences in the impact of location on the quality of varieties?
7) Fig.5. Shouldn't the RDA be for 2 years together? This would take into account the full range and variability of temperatures and other features.
8) Fig. 6. Edit error. Fig 6 is actually Fig 5b. There are "Q" symbols on the figure and "T" symbols in the header. In figures 5 and 6 there are different explanations for the Y9 symbol. Is the variable "Y9" the weight of fruit or FNPBS ? Furthermore, the distances between the letters and special characters need to be corrected.
9) Fig 5 and Fig 6 I suggest you enter the full name of vitamin C for the variable Y2 (=vitamin C)
10) The discussion is not very convincing and does not satisfactorily explain the results obtained. This is generally "Results 2" (except for the nutrients function). The results are well described, so you can try to explain more fully the differences between plants and varieties. Was there a "genotype x environment" interaction? Which varieties to choose for specific weather and/or soil conditions.
11) Soil properties: description of chemical methods insufficient, too laconic. What is important for the reader is the method used, a short description and the analytical equipment used. The reference method number is not enough. It also applies to the assessment of fruit quality.
12) The description of statistical procedures is also incomplete. Was there an analysis of variance before Duncan's test?
13) Conclusions - needs to be improved. There are no practical recommendations - what varieties to grow on soils poor in nutrients and/or in different climatic zones.
14) Supplementary Materials - In subsection, full titles of tables and figures included in the supplementary material should be provided.
15) The list of literature needs to be improved.
Author Response
R1: The text should be editorially improved. Latin names should be written in italics, e.g. line 15 and other places in the text. The space between letters and special characters is often not preserved, e.g. lines 16, 24, 80, 204 and other places in the text.
A1:Thank you for your kindly suggestion. We have unified the space between letters, special characters and lines in the whole revised manuscript.
R2:The scientific notation of the units is as follows: g·kg-1 not g/kg
A2: The scientific notation of the units has been changed to g·kg-1.
R3: There is no fruit weight in 2022, why
A3:Because the calculation of weighing was not performed owning to the Covid 19 in the second year. However, the fruit weight didn’t change a lot among years.
R4:There are no full statistics in Table 4. There is a coefficient of variation, but how to compare varieties?
A4: We have separated Table 4 into two tables ie. newly Table 4 and Table 5. And added statistical parameters among different cultivars.
R5: I suggest unifying the terms for the location, currently the names are double, and additionally they are used interchangeably, for example Fuping once and then Hebei (line 141-159). This will make it easier for the reader to perceive the data.
A5: We have unified the location into Fuping(Hebei), Taigu (Shanxi), Alar (Xinjiang) respectively in all the revised manuscript.
R6:The description of Fig. 3 should be complete and clearer. Mark clearly what is a location and what is a variety. It is necessary to explain the letters on the chart and the error bars. Figures can be presented in color. There is also no explanation of how to compare the results. Do we assess differences in the impact of location on the quality of varieties?
A6: We have revised the notes under Fig. 3 and make all the abbreviations clear. And also, we changed the Fig. in color. We described the impact of cultivars, regions and years on the tested nutritions in the revised manucript (Line 209-288).
R7:Fig.5. Shouldn't the RDA be for 2 years together? This would take into account the full range and variability of temperatures and other features.
A7: Very good suggstion. The RDA was for 2 years together and we merged them together.
R8: Fig. 6. Edit error. Fig 6 is actually Fig 5b. There are "Q" symbols on the figure and "T" symbols in the header. In figures 5 and 6 there are different explanations for the Y9 symbol. Is the variable "Y9" the weight of fruit or FNPBS ? Furthermore, the distances between the letters and special characters need to be corrected.
A8: Acturally, Fig. 6 was dfferent with Fig. 5. The former showed the relationship between jujube fruit traits and soil environmental variables while the latter showed the relationship between jujube fruit characteristics and climate variables. The distances between the letters and special characters have be corrected in the whole manuscript.
R9:Fig 5 and Fig 6 I suggest you enter the full name of vitamin C for the variable Y2 (=vitamin C)
A9: I accepted this suggestion. The symbols of the paper has been revised as suggested in Fig. and Fig. 6.
R10: The discussion is not very convincing and does not satisfactorily explain the results obtained. This is generally "Results 2" (except for the nutrients function). The results are well described, so you can try to explain more fully the differences between plants and varieties. Was there a "genotype x environment" interaction? Which varieties to choose for specific weather and/or soil conditions.
A10: Very constructive suggestion. We have revised in the Discussion part.
R11: Soil properties: description of chemical methods insufficient, too laconic. What is important for the reader is the method used, a short description and the analytical equipment used. The reference method number is not enough. It also applies to the assessment of fruit quality.
A11:We have added the description of chemical methods, analytical equipment etc.(Line 470-478).
R12: The description of statistical procedures is also incomplete. Was there an analysis of variance before Duncan's test?
A12: We have added one-way analysis of variance before Duncan’s test.And also, principal component analysis was also described (Line 507-510).
R13: Conclusions - needs to be improved. There are no practical recommendations - what varieties to grow on soils poor in nutrients and/or in different climatic zones.
A13: In view of the productivity and fruit quality, Yuhong hehaved better in Fuping (Hebei) and Zaocuimi, Zaoqiuhong, Fucuimi were suitable to be cultivated in Taigu (Shanxi). In Alar (Xinjiang), more cultivar such as Zaocuimi, Yuhong, Yulu, Luzao 2 and Yueguang showed promising future (Line 520-523).
R14: Supplementary Materials - In subsection, full titles of tables and figures included in the supplementary material should be provided.
A14: I accepted this suggestion. And we have revised the full titles of tables and figures.
R15: The list of literature needs to be improved
A15: We have carefully checked and improved the format of the all the literature according to the request of the Plants.
Reviewer 2 Report
Comments and Suggestions for Authors
The manuscript needs to be rewritten and better structured. Now, it is difficult to read and understand the main line, because of in some parts of overcrowded text, in some parts of lacking information, in some parts not correct presentation of data.
Also, many style corrections should be made in the text. One of examples: ‘In 2021, the ratio of sugar to acid and the ratio of total soluble solid to acid are important indexes to evaluate the taste and flavor of fruit.’ Does it mean that in 2022 or other years it will be not important? Another example: ‘’Lengbaiyu and Luzao 2 were higher in Fuping and Alar, while Yuhong and Yulu were significantly different in the three test sites.’’ What was higher?
Avoid redundant and repeating information as in section 2.6. Relationship between jujube fruit quality and meteorological factors: line 238-241: ’’The data …. from June to September 2021 were selected for redundancy analysis (RDA)’’ and line 256-258: ‘’In order to further verify the influence …..were selected for redundancy analysis (RDA)’’.
Very difficult to follow the text when there is a mess in the indication of sites. From the beginning sites in the text are Shanxi, Xinjiang and Hebei, but in the Fig.1 and 2 Fuping, Alar, Taigu are indicated. Later in the text Fuping, Alar, Taigu appear but in the figures 3 and 4 Shanxi, Xinjiang and Hebei are indicated.
From the Results part many overall statements or explanations should be removed to the Discussions. Meanwhile, Discussion part lacks discussions – all references provided only in paragraph one and nothing in the rest of the text.
Statistical evaluation is not provided in the tables 2-4. Much more statistical interpretations are needed:
statistics in figures indicate differences between sites but not between cultivars and not between years. What is the site cultivar interaction? What is year effect on the cultivar performance?
Some mistakes are in the figures and not correct interpretations in the text:
Fig.1. Two graphs Average temperature and mean high temperature are the same.
Not correct statement ‘’ In 2021, the monthly average temperature of the three test sites was the highest in July...’’ According to Fig.1 in Fuping the highest was in June.
Table 2. make the same order of cultivars in the different regions. It will be easier to follow the text and compare cultivars.
Why is the phenological data from 2022 not provided?
Table 3. Indicate year.
Table 4 Why there are no fruit weight data from 2022?
Fig.3 and 4. Does the ratio (sugar/acid is evaluated in percents?
In Materials and Methods you indicate that ‘’Jujube fruits were collected in 2021’’, but in the manuscript you provide fruit quality data from 2022 too.
Analysis of the relationships must be the hart of the manuscript, meanwhile it gives more questions than answers. 2.6. Relationship between jujube fruit quality and meteorological factors. Here you provide relationships separately for 2021 and 2022. More detailed discussion are needed what are the differences between years, what are interactions, what is the role of genotype.
2.7. Relationship between jujube fruit quality and soil conditions. All characteristics were analysed in two years. You did not provide differences between years, but it is obvious that cultivar/year and cultivar/site and site/year interactions are existing. How did you establish relationships? What year data did you use?
All these questions must be explained, and only then will be possible to make right conclusions. In presented version too many doubts arise.
And at the end mistakes are in the reference list too:
Repeated references 7 and 9. When one will be removed the citation order must be changed throughout the text.
Correct references: for example No3. ‘Liu MJ, wang JR, Liu p, Zhao J, Zhao ZH, et al.’; or No18. ‘Fect of potassium’.
There is a mess in the presentation of the references. Correct according to journal instructions.
Author Response
R1:The manuscript needs to be rewritten and better structured. Now, it is difficult to read and understand the main line, because of in some parts of overcrowded text, in some parts of lacking information, in some parts not correct presentation of data.
A1: Thank you for your kindly suggestion. We have carefully checked all the manuscript and improved the words, grammar and the structure.
R2:Also, many style corrections should be made in the text. One of examples: ‘In 2021, the ratio of sugar to acid and the ratio of total soluble solid to acid are important indexes to evaluate the taste and flavor of fruit.’ Does it mean that in 2022 or other years it will be not important? Another example: ‘’Lengbaiyu and Luzao 2 were higher in Fuping and Alar, while Yuhong and Yulu were significantly different in the three test sites.’’ What was higher?
A2: We have improved the description through all the manuscript.
R3:Avoid redundant and repeating information as in section 2.6. Relationship between jujube fruit quality and meteorological factors: line 238-241: ’’The data …. from June to September 2021 were selected for redundancy analysis (RDA)’’ and line 256-258: ‘’In order to further verify the influence …..were selected for redundancy analysis (RDA)’’.
A3: We have deleted the redundant and repeating information in the manuscript.
R4:Very difficult to follow the text when there is a mess in the indication of sites. From the beginning sites in the text are Shanxi, Xinjiang and Hebei, but in the Fig.1 and 2 Fuping, Alar, Taigu are indicated. Later in the text Fuping, Alar, Taigu appear but in the figures 3 and 4 Shanxi, Xinjiang and Hebei are indicated.
A4: The location identification were unified. They were changed to Fuping (Hebei), Taigu (Shanxi) and Alar (Xinjiang) respectively.
R5:From the Results part many overall statements or explanations should be removed to the Discussions. Meanwhile, Discussion part lacks discussions – all references provided only in paragraph one and nothing in the rest of the text.
A5: Some explanations of the Results section has been moved to the Discussion section. References have been added to the introduction and discussion sections.
R6:Statistical evaluation is not provided in the tables 2-4. Much more statistical interpretations are needed: statistics in figures indicate differences between sites but not between cultivars and not between years. What is the site cultivar interaction? What is year effect on the cultivar performance?
A6: The main phenological period including bud sprouting, early flowering, full flowering, fruit crisp mature, fruit full mature, defoliation were recorded in table 2. Fruit development duration and annual growth duration were also sumerized in Table 2. In Table 3 and Table 4, we have added the standard deviation and significance test. And also, the differences between cultivars and between years were explained.
R7:Some mistakes are in the figures and not correct interpretations in the text:Fig.1. Two graphs Average temperature and mean high temperature are the same.Not correct statement ‘In 2021, the monthly average temperature of the three test sites was the highest in July...’’ According to Fig.1 in Fuping the highest was in June.
A7:We have checked and corrected the graphs.
R8:Table 2. make the same order of cultivars in the different regions. It will be easier to follow the text and compare cultivars.
A8:Good suggestion. We arranged the cultivar order by the fruit development duration in the same location. However, we have reordered by initial alphabet of cultivar names.
R9:Why is the phenological data from 2022 not provided? Table 3. Indicate year. Table 4 Why there are no fruit weight data from 2022?
A9:Unfortunately, the phenological period and the fruit weight was not observed owning to the Covid 19 in 2022.
R10:Fig. 3 and 4. Does the ratio (sugar/acid is evaluated in percents? In Materials and Methods you indicate that ‘’Jujube fruits were collected in 2021’’, but in the manuscript you provide fruit quality data from 2022 too.
A10:Acturally, sugar/acid is a value instead of percent. In 2022, the fruits were picked, sliced and stored under liquid nitrogen for further nutrition detection. We have explained the method in detail in Materials and Methods.
R11:Analysis of the relationships must be the hart of the manuscript, meanwhile it gives more questions than answers. 2.6. Relationship between jujube fruit quality and meteorological factors. Here you provide relationships separately for 2021 and 2022. More detailed discussion are needed what are the differences between years, what are interactions, what is the role of genotype.
2.7. Relationship between jujube fruit quality and soil conditions. All characteristics were analysed in two years. You did not provide differences between years, but it is obvious that cultivar/year and cultivar/site and site/year interactions are existing. How did you establish relationships? What year data did you use?All these questions must be explained, and only then will be possible to make right conclusions. In presented version too many doubts arise.
A11: Through further analyses in Part 2.5 Influences of environmental conditions on fruit quality, we found that all the fruit quality parameters were differed with locations, cultivars and years. However, it’s difficult to say which factor is the most important factor. So, we employ redundancy analysis (RDA) and try to expain in large scale by using 2 years data in 2.6. Relationship between jujube fruit quality and meteorological factors and 2.7 Relationship between jujube fruit quality and soil conditions.
R12:And at the end mistakes are in the reference list too:
Repeated references 7 and 9. When one will be removed the citation order must be changed throughout the text.
Correct references: for example No3. ‘Liu MJ, wang JR, Liu p, Zhao J, Zhao ZH, et al.’; or No18. ‘Fect of potassium’.
There is a mess in the presentation of the references. Correct according to journal instructions.
A12:We have checked all the references one by one and corrected them.
Reviewer 3 Report
Comments and Suggestions for Authors
ID: plants-2688308
Main comments to authors
1. The abstract needs to be corrected. Instead of a research hypothesis, there is a long-winded introduction. There is no final conclusion from the study.
A correctly constructed abstract contains: research hypothesis, basic elements of the methodology, basic results and the final conclusion that summarizes the research. The length of the abstract is of 200 words.
2. A very poor introduction. In addition, paragraphs 1 and 3 are not supported by the cited scientific literature.
3. The presented results are the result of two years of field studies conducted in three locations. Such results require a synthesis in which, years are a factor. Conclusion cannot be drawn without synthesis. Only after synthesis will this study have scientific value.
Author Response
R1: The abstract needs to be corrected. Instead of a research hypothesis, there is a long-winded introduction. There is no final conclusion from the study.
A1: Thank you for your suggestion. We have revised the abstract to add the final conclusions of this study.
R2: A correctly constructed abstract contains: research hypothesis, basic elements of the methodology, basic results and the final conclusion that summarizes the research. The length of the abstract is of 200 words.
A2: The abstract has been revised and the number of words meets the requirements.
R3: A very poor introduction. In addition, paragraphs 1 and 3 are not supported by the cited scientific literature.
A3: We have revised the introduction and added new literature to support in Line 49-67.
R4: The presented results are the result of two years of field studies conducted in three locations. Such results require a synthesis in which, years are a factor. Conclusion cannot be drawn without synthesis. Only after synthesis will this study have scientific value.
A4: We have carefully analyzed the results deeply and hope meet the request of the reviewers.
Reviewer 4 Report
Comments and Suggestions for Authors
Comments and remarks:
2.1. Differences of meteorological factors and soil conditions between test sites in different ecological regions
You wrote here about meteorological factors in provinces, but the results of measurement in the figures 1 and 2 are only for Fuping, Taigu and Alar (or country/district), not the provinces. A better solution is Fuping (Hebei), Taigu (Shanxi) and Alar (Xinjiang) or Fuping in Hebei, Taigu in Shanxi and Alar in Xinjiang or Fuping/Hebei, etc.
2.2 Influences of environmental conditions on phenological periods
Based on the research performed in Fuping, Taigu and Alar, it is not possible to describe the results in the Hebei, Shanxi and Xinjiang provinces. Here you should only write about Fuping (Hebei), Taigu (Shanxi) and Alar (Xinjiang) or ... look at the comment above. Correction required
In the description of the phenological periods for plants growth in Hebei and Alar you usually provide dates, e.g. from April 18th (Yueguang) to April 21st, while in the description of Taigu you also provide the duration of the phenological period, e.g. 3 days. This needs to be standardised.
There is no explanation why in Table 2 the order of jujube varieties is different in each location. According to what criterion was this order determined? According to fruit development duration?
2.4. Influences of environmental conditions on productivity
Comments on the research location as above. Correction required
You wrote:
“According to the investigation in 2021, the averaged FNPBS of 8 cultivars in Xinjiang is significantly higher than that of Hebei, and the averaged …..”
It is not known whether the differences are significant because there are no statistical calculations. It should be removed "significantly".
In table 4 the coefficient of variation calculated for FNPBS and fruit weight is very large. The higher the coefficient of variation, the greater the level of dispersion around the mean. It means that standard deviation is relatively large compared to the mean. In my opinion, the results are too varied to be considered reliable.
2.5. Influences of environmental conditions on fruit quality
You wrote:
“In 2021, there was little difference in titratable acid content among the three sites, indicating that environmental changes had little effect on the titratable acid content. But there was a great difference among cultivars in the same site (Fig 3C). The titratable acid content of Yulu was significantly higher than that of other cultivars and topped 1.99% in Taigu.”
It is not known whether it is significant because there are no statistical calculations related to differences between varieties of jujube. These are only calculations for jujube growing locations. It should be removed "significantly"
2.5. Influences of environmental conditions on fruit quality
Fig. 3 and Fig. 4 are described with province symbols, and the research sites should be Fuping, Taigu and Alar (see Fig. 1 and 2).
In whole chapter 2.5. The statistical calculation applies to differences between jujube growing locations. There are no statistical calculations in this publication regarding differences between varieties. Therefore, it cannot be said that the varieties differed significantly.
You can write that something is bigger/larger or smaller, but you cannot write about the significance of the differences.
Correction required.
Fig. 5. is the relationship between jujube fruit characteristics and climate variables based on CANOCO software RDA analysis in 2021 (A) and 2022 (B).
Line 241 and 245:
The FNPBS was correlated….
Were on Fig. 5 is FNPBS (Fruit number per bearing shoot) ?
2.7. Relationship between jujube fruit quality and soil conditions
Line 276 and 280:
Were on Fig. 6 is FNPBS ?
Fig. 6
In the figure there are fruit characteristics Y1, Y2 etc. and Q1,Q2 etc. What do the letters Q mean?
Under Fig. 6 (line 296 – 300) soil variables marked as T.
Symbols should be unified.
Author Response
R1: Differences of meteorological factors and soil conditions between test sites in different ecological regions
You wrote here about meteorological factors in provinces, but the results of measurement in the figures 1 and 2 are only for Fuping, Taigu and Alar (or country/district), not the provinces. A better solution is Fuping (Hebei), Taigu (Shanxi) and Alar (Xinjiang) or Fuping in Hebei, Taigu in Shanxi and Alar in Xinjiang or Fuping/Hebei, etc.
A1: Very kind suggestion. I have changed them to Fuping (Hebei), Taigu (Shanxi) and Alar (Xinjiang)
R2: Influences of environmental conditions on phenological periods
Based on the research performed in Fuping, Taigu and Alar, it is not possible to describe the results in the Hebei, Shanxi and Xinjiang provinces. Here you should only write about Fuping (Hebei), Taigu (Shanxi) and Alar (Xinjiang) or ... look at the comment above. Correction required.
In the description of the phenological periods for plants growth in Hebei and Alar you usually provide dates, e.g. from April 18th (Yueguang) to April 21st, while in the description of Taigu you also provide the duration of the phenological period, e.g. 3 days. This needs to be standardised.
There is no explanation why in Table 2 the order of jujube varieties is different in each location. According to what criterion was this order determined? According to fruit development duration?
A2: I have changed them to Fuping (Hebei), Taigu (Shanxi) and Alar (Xinjiang). As to the description of the phenological periods, we have standardised.
Yes, in Table 2, we arranged the cultivars according to fruit development duration in each location. Meanwhile, we have re-arranged the cultivar by initial alphabet.
R3: Influences of environmental conditions on productivity
Comments on the research location as above. Correction required
You wrote:“According to the investigation in 2021, the averaged FNPBS of 8 cultivars in Xinjiang is significantly higher than that of Hebei, and the averaged …..”. It is not known whether the differences are significant because there are no statistical calculations. It should be removed "significantly".In table 4 the coefficient of variation calculated for FNPBS and fruit weight is very large. The higher the coefficient of variation, the greater the level of dispersion around the mean. It means that standard deviation is relatively large compared to the mean. In my opinion, the results are too varied to be considered reliable.
A3: All the results were observed under the normal and same cultivating condition. The large variation between regions and years, probably indicated that the meteorological and soil conditions affected the development the the tree and fruits.
R4: Influences of environmental conditions on fruit quality
You wrote:“In 2021, there was little difference in titratable acid content among the three sites, indicating that environmental changes had little effect on the titratable acid content. But there was a great difference among cultivars in the same site (Fig 3C). The titratable acid content of Yulu was significantly higher than that of other cultivars and topped 1.99% in Taigu.” It is not known whether it is significant because there are no statistical calculations related to differences between varieties of jujube. These are only calculations for jujube growing locations. It should be removed "significantly"
A4: I have removed "significantly".
R5: Influences of environmental conditions on fruit quality
Fig. 3 and Fig. 4 are described with province symbols, and the research sites should be Fuping, Taigu and Alar (see Fig. 1 and 2).
In whole chapter 2.5. The statistical calculation applies to differences between jujube growing locations. There are no statistical calculations in this publication regarding differences between varieties. Therefore, it cannot be said that the varieties differed significantly.
You can write that something is bigger/larger or smaller, but you cannot write about the significance of the differences.
Correction required.
Fig. 5. is the relationship between jujube fruit characteristics and climate variables based on CANOCO software RDA analysis in 2021 (A) and 2022 (B).
Line 241 and 245:
The FNPBS was correlated….
Were on Fig. 5 is FNPBS (Fruit number per bearing shoot) ?
A5: Good suggestion. We have unified modification to Fuping(Hebei), Taigu (Shanxi), Alar (Xinjiang) and removed "significantly". Fig. 5 showed the relationship between jujube fruit characteristics and climate variables. Fruit characteristics included total soluble solid, titratable acid, soluble sugar,ratio of total soluble solid to acid, ratio of sugar to acid, fruit number per bearing shoot, total phenolsm, total flavonoids, fruit weight.
R6: Relationship between jujube fruit quality and soil conditions
Line 276 and 280:
Were on Fig. 6 is FNPBS ?
Fig. 6
In the figure there are fruit characteristics Y1, Y2 etc. and Q1,Q2 etc. What do the letters Q mean?
Under Fig. 6 (line 296 – 300) soil variables marked as T.
A6: We have revised Fig.6 in Line 346-353 and make every symbol clear.
Reviewer 5 Report
Comments and Suggestions for Authors
The goal of the paper is to present the correlation between the fruit contents and the climatic factors and soils so the data needs to be presented in a different way. i.e. Fig meteo factors should be in one table and Statistic of the correlations should be presented: the correlation tables with formula, significancy , type of correlation, etc.
the descriptive tables and graphs can be in the link.
Author Response
R1:The goal of the paper is to present the correlation between the fruit contents and the climatic factors and soils so the data needs to be presented in a different way. i.e. Fig meteo factors should be in one table and Statistic of the correlations should be presented: the correlation tables with formula, significancy, type of correlation, etc.
the descriptive tables and graphs can be in the link.
A1: Thank you for your kindly suggestion. However, the meteo factors will be more easier to know the trends among month. So, we employ Figure instead of table to show the meteo factors. and also, the relationship between jujube fruit characteristics and climate variables analyzed by CANOCO software RDA was more clear to dispalyed in figure. In addition, we have explained how to explain the figure in Line 308-312. If needed, we can submit the the correlation tables and descriptive tables as attachment.
Round 2
Reviewer 1 Report
Comments and Suggestions for Authors
The manuscript has been revised in accordance with the comments and suggestions included in the review. The authors put a lot of work into improving both the substantive and editorial comments. Despite this, the manuscript still requires improvement. However, these are editorial comments.
1. Methods for determining nutrients are still poorly described. The authors added the equipment and analytical method, but did not provide the exact name of the equipment, company and country. The most important thing is that there is no information about the solutions in which the nutrients were determined. The authors listed the method numbers (e.g. NY/T1121.2-2006), but the reader still does not know whether the soil pH was determined in 1 M KCl, 0.01 M CaCl2 or other solutions! Each country has different method symbols. The essence of the method is important.
2. Please specify clearly which method was used: RDA or PCA. These are different methods. Currently, in some places of the text it is RDA and in others it is PCA.
3. when two article numbers are given, a comma is used, not a dash. Example: Page 2, Line 60: it is [16-17] and it should be [16,17]
4. In the literature list, the issue number should be written in italics
5. My earlier comment no. 14 is not taken into account, despite the positive response from the authors. The Supplementary Material section should include the titles of tables and figures. Please see other articles published in MDPI journals.
Reviewer 2 Report
Comments and Suggestions for Authors
Though authors declared that made all suggested corrections and changes (and they did some changes indeed), still there are a lot of mistakes left and new mistakes done.
Lines 42, 45. reference number 0?
Line 45 is it correct that 8 references are used to support one statement [0-8]? Again, reference number 0.
Line 396 Is it correct that 33 sources are used in one sentence [0-33]? And again, reference number 0.
Lines 414-416. Provided references 38 and 39 do not support your findings as they are not presenting correlations with mineral content.
References 10 and 38 are the same.
So, rhetorical question: If there is such mess in such simple task, what can reviewer expect from your provided results and interpretations? Should we recount averages, as for example in Table 3 with Zaoqiuhong cultivar?
Reviewer 3 Report
Comments and Suggestions for Authors
The manuscript has been significantly improved and I accept it as is.
Author Response
请参阅附件。(Please see the attachment.)

Round 3
Reviewer 2 Report
Comments and Suggestions for Authors
The third review:
1. Table 5. Please, correct letters indicating significant differences in Fuping site: Yulu cannot be marked by ‘c’.
2. Fig 4 H. Check statistical evaluation. Values differ several times, but not significant differences found?
3. In introduction and discussion parts the note appears ‘’[Error! Reference source not found’’
4. I am spending the third time at your manuscript, writing the same remarks: now new references - 15 and 23 - are the same!!! It means you have to change the order of references in the text.
Round 4
Reviewer 2 Report
Comments and Suggestions for Authors
According to your statement “In addition, the characteristics of jujube fruit are significantly related to temperature. High temperature significantly affected the accumulation of sugar, degradation of organic acid content, accumulation of anthocyanins and skin coloring [21-24]. However, the specific effects of ecological conditions on the growth and development of different jujube cultivars remained unclear.”, all references should present research on jujube. But you use not correct references: no 21 is about poncan, no 23- grapes, no 24 – mandarin.
According to journal rules all references should be numbered according to the order of their appearance in the text. Now, in Introduction references are 1-24, references in Discussions 30 - 43, but in the last MM section numbering of references are in between 25-29.
